# A 3D basicranial shape-based assessment of local and continental northwest European ancestry among 5th to 9th century CE Anglo-Saxons

Kimberly A. Plomp[1,2]*, Keith Dobney[1,2,3], Mark Collard[1]*

1 Department of Archaeology, Simon Fraser University, Burnaby, British Columbia, Canada, 2 Department of Archaeology, Classics, Egyptology, University of Liverpool, Liverpool, United Kingdom, 3 School of Historical and Philosophical Inquiry, University of Sydney, Sydney, NSW, Australia

* kplomp@sfu.ca (KAP); mcollard@sfu.ca (MC)

## Abstract

The settlement of Great Britain by Germanic-speaking people from continental northwest Europe in the Early Medieval period (early 5th to mid 11th centuries CE) has long been recognised as an important event, but uncertainty remains about the number of settlers and the nature of their relationship with the preexisting inhabitants of the island. In the study reported here, we sought to shed light on these issues by using 3D shape analysis techniques to compare the cranial bases of Anglo-Saxon skeletons to those of skeletons from Great Britain that pre-date the Early Medieval period and skeletons from Denmark that date to the Iron Age. Analyses that focused on Early Anglo-Saxon skeletons indicated that between two-thirds and three-quarters of Anglo-Saxon individuals were of continental northwest Europe ancestry, while between a quarter and one-third were of local ancestry. In contrast, analyses that focused on Middle Anglo-Saxon skeletons suggested that 50–70% were of local ancestry, while 30–50% were of continental northwest Europe ancestry. Our study suggests, therefore, that ancestry in Early Medieval Britain was similar to what it is today—mixed and mutable.

## Introduction

The settlement of large parts of the island of Great Britain by Germanic-speaking people from continental northwest Europe between the mid 5th and early 7th centuries CE has long been recognised as an important event, leading as it did to the formation of the ethnic group called the Angli or, more commonly, the Anglo-Saxons [1]; the development of the English language; and the formation of the Kingdom of England and, eventually, the United Kingdom [2–5]. Not surprisingly, therefore, this episode has been the subject of a considerable amount of research (see Hardland [6] for a recent review). Despite this extensive work, uncertainty remains about the number of settlers and the nature of their relationship with the preexisting inhabitants of the island, especially the Romano-British.

**Data Availability Statement:** All relevant data are within the manuscript and its Supporting Information files.

1 / 15

**Funding:** The study was supported by the European Union's Marie Skłodowska-Curie Actions program (Horizon 2020 - 748200), the Social Sciences and Humanities Research Council of Canada (895-2011-1009), the Canada Research Chairs Program (228117 and 231256), the Canada Foundation for Innovation (203808), the British Columbia Knowledge Development Fund (862-804231), and Simon Fraser University (14518).

**Competing interests:** The authors have declared that no competing interests exist.

Traditionally, knowledge of the settlement of Britain by the Angles, Saxons, Jutes, and Frisians relied on two historical texts, *Historia Ecclesiastica Gentis Anglorum* and the *Anglo-Saxon Chronicle*. Written by the Venerable Bede, *Historia Ecclesiastica* is thought to have been completed in 731 CE. The original version of the *Anglo-Saxon Chronicle* was compiled in the late 9th century CE, and copies of it were updated until at least the mid 12th century CE. Both of these documents describe a mass invasion and a rapid replacement of the indigenous population [7–9].

This picture has been challenged by archaeologists. Several researchers have argued that the archaeological record reveals that the changes associated with the arrival of the Germanic-speaking settlers in Britain occurred relatively slowly and that this is inconsistent with the idea that the settlers replaced the Romano-British [10–12]. In line with this, analyses of oxygen and strontium isotopes from Anglo-Saxon skeletons have found that only a small minority of the sampled individuals were from the Continent [13–15].

Geneticists have also sought to shed light on these issues, but the results they have obtained are highly variable. A comparison of the ancestry estimates reported by Weale et al. [3], Leslie et al. [16], and Schiffels et al. [17] illustrates this. Based on analyses of modern Y chromosome DNA, Weale et al. [3] concluded that 50–100% of men in central England have male ancestors from continental northwest Europe. Leslie et al.'s [16] analyses of modern genomes suggested that 10–40% of people from central and southern England have continental northwest European ancestry. Schiffels et al. [17] analysed the genomes of ten Iron Age and Anglo-Saxon individuals from England. They found that the Anglo-Saxon individuals were closely related to modern Danish and Dutch people and estimated that introgression from continental northwest Europe accounted for 38% of the ancestry of people currently living in eastern England.

Here, we report a study designed to bring a new line of evidence to bear on the issue. We used three-dimensional (3D) shape analysis techniques to compare the cranial bases of Anglo-Saxon skeletons to those of skeletons from Great Britain that pre-date the Early Medieval period and skeletons from Denmark that date to the Iron Age. We focused on the basicranium because previous studies have shown that the 3D shape of this region of the skull can be informative about relatedness among human populations (e.g. [18–21]). The shape analysis techniques we utilised are collectively referred to as 'Geometric Morphometrics' [22–24]. These techniques have been used extensively by palaeoanthropologists to tackle comparable problems (e.g. [25, 26]). The goal of the analyses was to estimate the percentage of Anglo-Saxon individuals who were of British ancestry and the percentage who were of continental northwest European ancestry.

## Materials and methods

The sample is summarised in Table 1 and the locations of the sites from which the remains were obtained are shown in Fig 1. Further information about the individuals in the sample can be found in the S1 File.

We recorded data on a total of 236 individuals, all of whom had an intact basicranium. Only adults were measured to avoid the confounding effects of ontogeny; individuals were

**Table 1. Groups used in the study.**

| Group name | Approximate date range | ♀ | ♂ |
|---|---|---|---|
| Early Anglo-Saxons | 410 CE-660 CE | 27 | 20 |
| Middle Anglo-Saxons | 660 CE-899 CE | 13 | 29 |
| Pre-Medieval British | 800 BCE-410 CE | 45 | 56 |
| Danish | 800 BCE-399 CE | 14 | 32 |

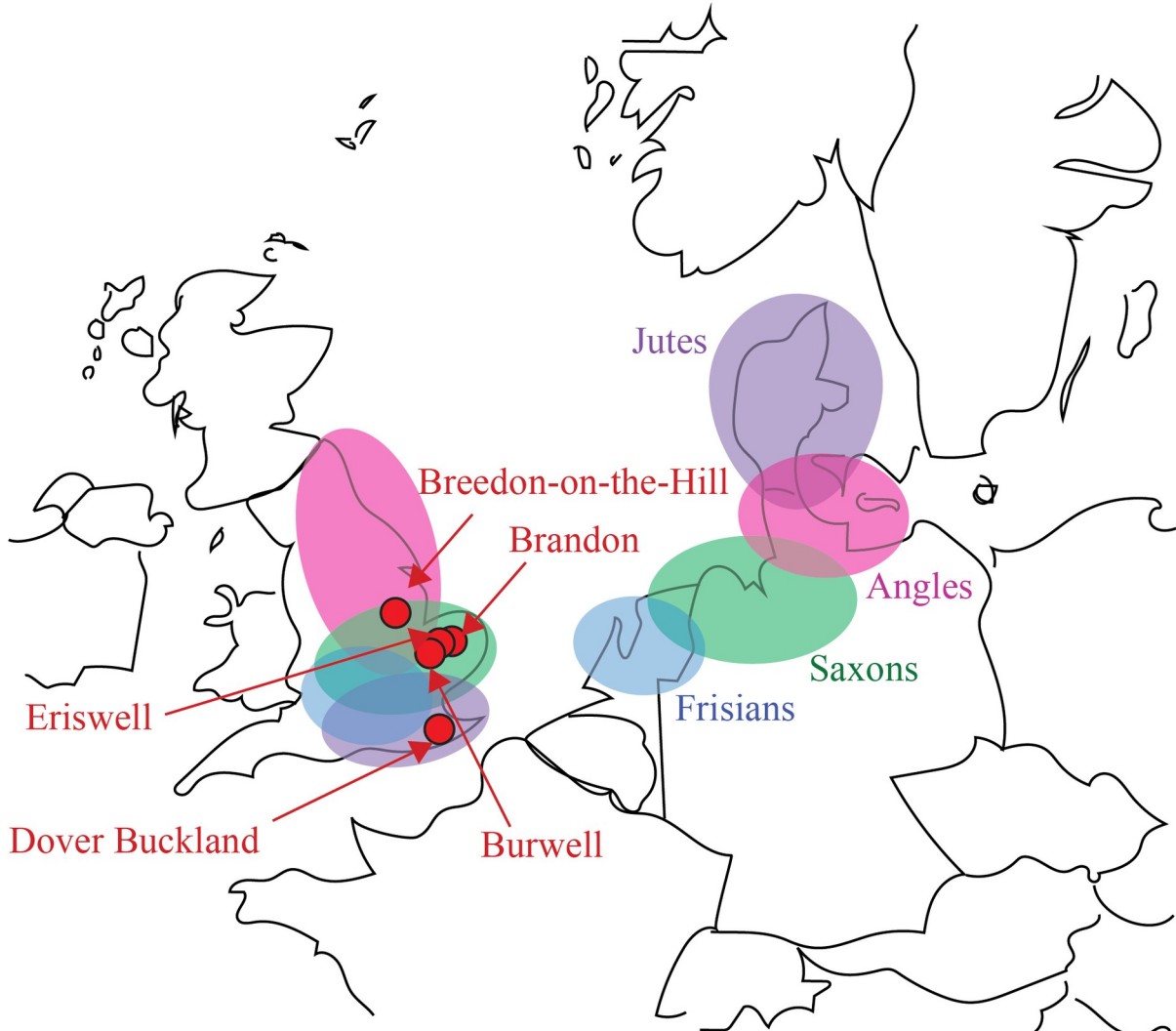

**Fig 1. Map depicting the locations of the five Anglo-Saxon cemeteries sampled in this study (red dots).** The map also shows the homelands and destinations of the main Germanic-speaking groups who settled in Britain in the Early Medieval period—the Angles (pink ellipses), Saxons (green ellipses), Jutes (purple ellipses), and Frisians (blue).

judged to be adult on the basis of dental eruption and epiphyseal fusion. Both males and females were included in the sample, with sex being estimated primarily on the basis of pelvic morphology, especially the presence/absence of the ventral arc [27]. Cranial indicators of sex were also considered when necessary [27].

Eighty-nine of the individuals come from five Anglo-Saxon cemeteries: Brandon in Suffolk, Breedon-on-the-Hill in Leicestershire, Burwell in Cambridgeshire, Dover Buckland in Kent, and Eriswell in Suffolk. Breedon-on-the-Hill, Dover Buckland, and Eriswell date to the Early Anglo-Saxon Period (410 CE-660 CE), while Brandon and Burwell date to the Middle Anglo-Saxon Period (660 CE-889 CE). Hereinafter, we will refer to the individuals from Breedon-on-the-Hill, Dover Buckland, and Eriswell as Early Anglo-Saxons, and the individuals from Brandon and Burwell as Middle Anglo-Saxons.

Another 101 of the individuals are from pre-Medieval sites in England. The sites in question are Poundbury and Maiden Castle in Dorset, and Hallet's Garage in Kent. Poundbury and

Hallet's Garage are both of Romano-British date (43 CE-410 CE), while Maiden Castle dates to the preceding Iron Age (800 BCE-43 CE).

The remaining 46 individuals are from various sites in Denmark dating to the Iron Age (800 BCE-399 CE). We focused on individuals from Denmark partly because Schiffels et al.'s [17] analyses indicated that the early English individuals in their sample from Cambridgeshire were genetically closer to modern Danish people, and partly because they are better preserved than similarly dated skeletal remains from the other main potential source areas, northern Germany and the Netherlands.

We used photogrammetry to generate 3D models of the crania. Each cranium was photographed 150 times with an eight-megapixel digital single-lens reflex Canon EOS 77D camera mounted with a Canon 50mm lens. The cranium was placed on a PalaeoPi rotating table and then photographs were shot at intervals of approximately 10˚, as per Evin et al. [28]. Subsequently, the photographs for a single cranium were converted into a 3D model in Metashape. The photographs were aligned at the 'high' accuracy level and 3D depth maps generated. After this, the 3D depth maps were used to create a mesh model of each cranium.

To digitise the 3D shape of the individuals' cranial bases, we imported the mesh models into MorphoDig [29] and recorded the 3D Cartesian coordinates of a total of 34 anatomical landmarks (Fig 2). The landmarks were chosen based on those used by Harvati and Weaver [18] to capture the shape of the basicranium. According to Bookstein's [30] widely used scheme, seven of the landmarks are Type I landmarks and 27 are Type II landmarks.

A single observer recorded the data (KAP). Intra-observer error was assessed in the manner outlined by Neubauer et al. [31, 32]. A single cranium was digitised ten times and then Morphologika [33] was used to compare the greatest Procrustes distance between the ten repeated landmark configurations with the smallest Procrustes distance between the non-repeated landmark configurations representing all the crania. The smallest distance between the non-repeated cranium was almost double the greatest distance between the repeated crania. According to Neubauer et al. [31, 32], this amount of error is unlikely to influence the analysis of the shape variance of the sample.

After assessing the intra-observer error, we divided the dataset by sex. We did so to avoid sexual dimorphism influencing the results.

We then sought to minimise the impact of a number of other potential confounding factors on each sex-specific dataset. Following Klingenberg et al. [33], we reflected and re-labelled the landmark coordinates and subjected the data to generalised Procrustes analysis, which removes translational and rotational effects and scales the configurations to centroid size. We removed asymmetry by calculating the average Procrustes coordinates between the original and reflected landmarks. These procedures were carried out in MorphoJ [33].

Having minimised the confounding effects of translation, rotation, size, and asymmetry, we divided the individuals in the female dataset into four groups: Early Anglo-Saxons, Middle Anglo-Saxons, Pre-Medieval British, and Danish. We then did the same for the male dataset.

Subsequently, we carried out three sets of analyses. In the first, we explored the shape variance among the four groups. We did this by applying canonical variates analysis (CVA) to the Procrustes coordinates. We conducted two analyses, one that concentrated on the female dataset and one that focused on the male dataset. The CVAs were performed in MorphoJ [33].

In the second set of analyses, we investigated whether there were significant shape differences among the four groups. The primary goal of these analyses was to ensure that the two potential source groups, the Pre-Medieval British and Danish, differed because this was a prerequisite for making meaningful statements about the ancestry of the Anglo-Saxon individuals. We first subjected each dataset to principal components analysis (PCA) followed by the principal component (PC) reduction procedure developed by Baylac and Frieβ [34]. This procedure

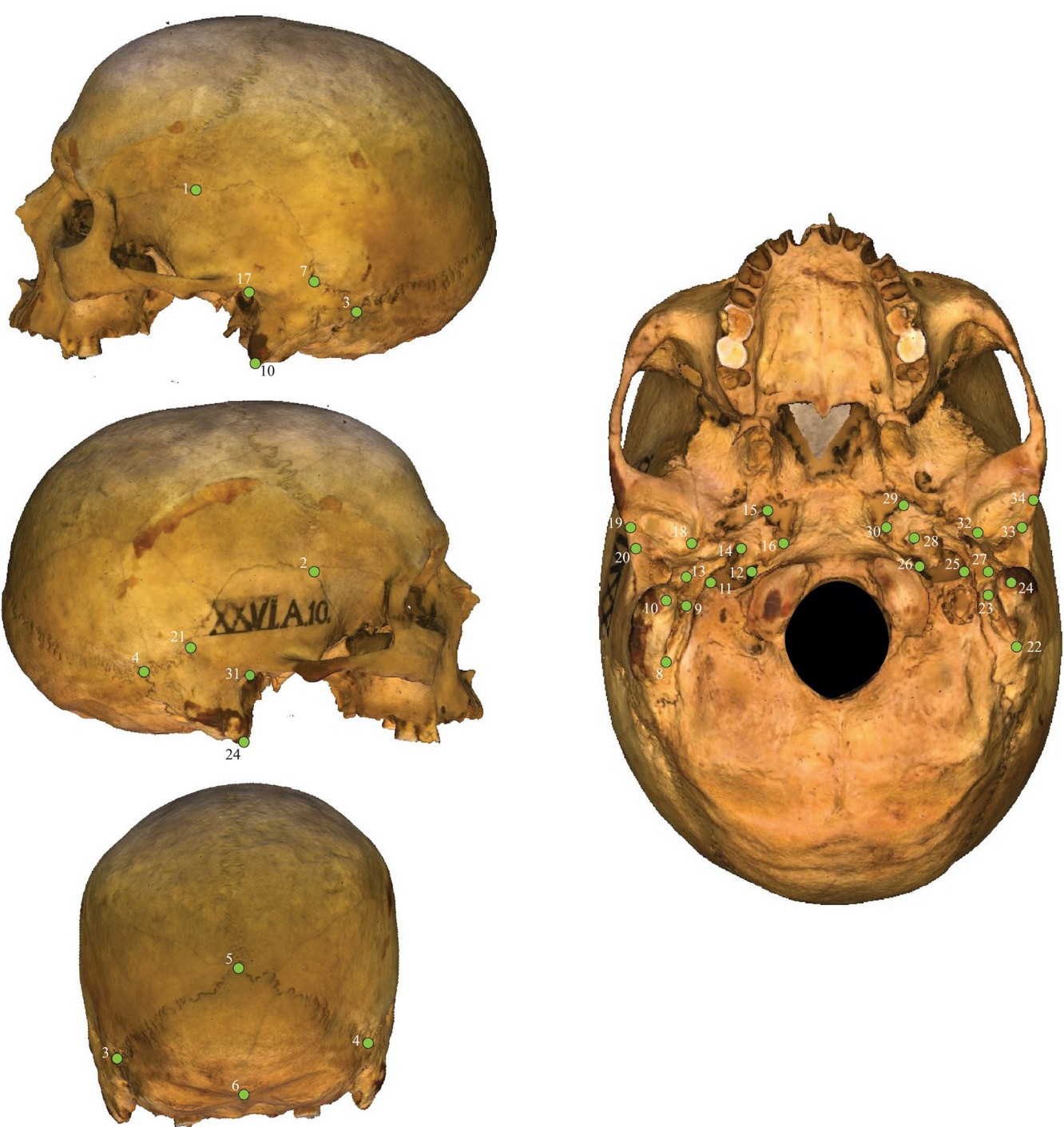

**Fig 2. Location of the 34 landmarks used in the present study.**

aims to reduce noise from PCs that account for little variance while still retaining all relevant shape information. It tackles this optimisation problem by progressively adding PCs into the analysis until the cross-validation percentage begins to drop. Thereafter, we subjected the retained PCs to MANOVA. Beginning with the females, we assessed whether there were any significant differences among the four groups. Because this analysis returned a significant

result, we proceeded to compare the populations on a pairwise basis, i.e. Danish vs Pre-Medieval British, Early Anglo-Saxons vs. Danish, Early Anglo-Saxons vs. Pre-Medieval British, Middle Anglo-Saxons vs. Danish, Middle Anglo-Saxons vs. Pre-Medieval British, Early Anglo-Saxons vs. Middle Anglo-Saxons. Subsequently, we repeated the MANOVAs with the males. The PCAs were performed in MorphoJ [33], while the MANOVAs were performed in R [35]. The PC-reduction procedure was implemented in R [35].

In the third set of analyses, we assessed the relative contribution of the two potential source groups to the ancestry of the two Anglo-Saxon groups. We did this by applying cross-validated linear discriminant analysis (LDA) to the PCs used in the previous set of analyses. Following Evin et al. [36], we designated the potential source groups as the known samples and then directed the LDA to indicate which of the potential source groups the Anglo-Saxon individuals most likely belonged. Because there are two potential source groups, the standard average percentage used to attribute an individual to one of the source groups is 50% [36, 37]. We opted for a more conservative affiliation value, and so an individual was deemed to be attributed to a given source group if the average percentage for that group was ≥55%. If both average attribution percentages for an individual were ≤54%, the individual was deemed to be unattributable. Thereafter, we calculated the percentage of Early Anglo-Saxons and Middle Anglo-Saxons that were attributed to each of the potential source groups. Once again, the female and male individuals were analysed separately. The LDAs were performed in R [35].

## Results

The CVA of the female individuals yielded three CVs. CV1 accounted for 58% of the variance, CV2 for 27%, and CV3 for 14%.

Fig 3A shows the values for CV1 and CV2 for the female individuals plotted against each other. The Pre-Medieval British and Danish overlap nearly completely on CV2 but only partially on CV1. On CV1, the bulk of the Pre-Medieval British individuals are located more positively than the majority of the Danish ones. The Early Anglo-Saxons overlap substantially with

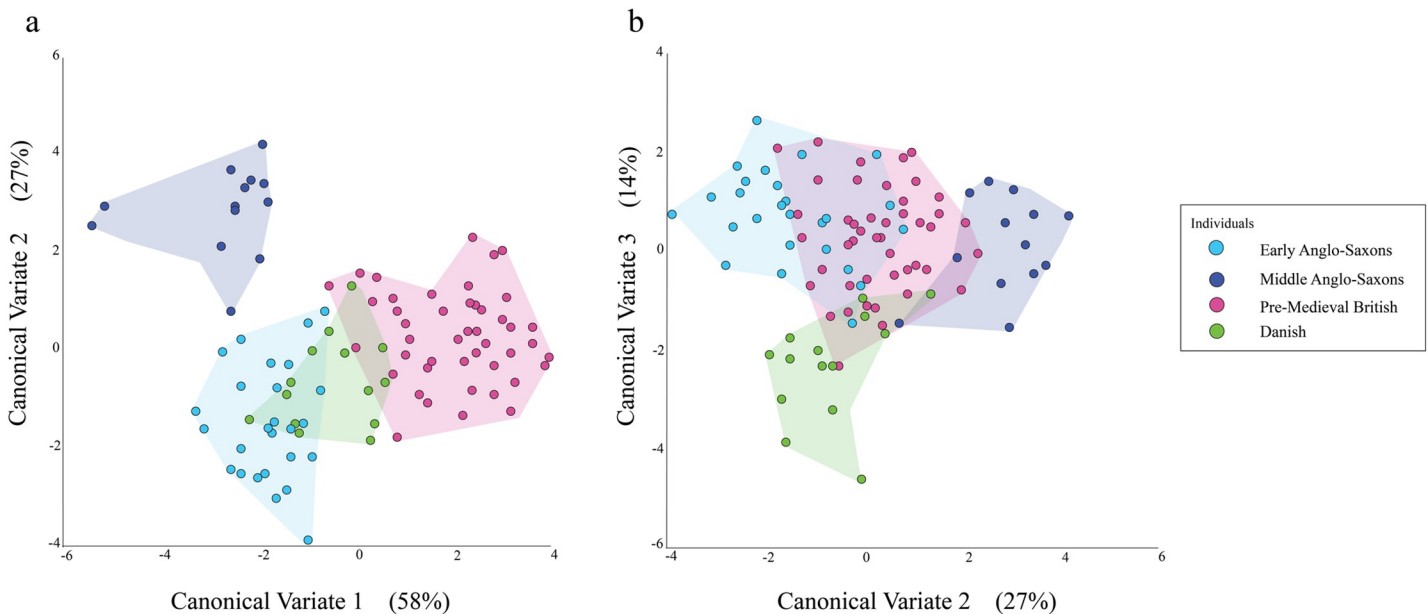

**Fig 3.** Plots of CV values for female individuals when (a) CV1 is plotted against CV2 and (b) CV2 is plotted against CV3.

both the Pre-Medieval British and the Danish on CV2, but only with the Danish on CV1. The Middle Anglo-Saxons overlap substantially with the Early Anglo-Saxons on CV1 but do not overlap with the Pre-Medieval British or Danish on the same CV. In contrast, the Middle Anglo-Saxons overlap somewhat with the Pre-Medieval British and Danish individuals on CV2; the overlap with the Pre-Medieval British individuals is greater than the overlap with the Danish ones.

The values for CV2 and CV3 for the females are plotted against each other in Fig 3B. The Pre-Medieval British and Danish individuals overlap almost entirely on CV2 but only a small amount on CV3. On CV3, the middle of the distribution of the Pre-Medieval British is closer to the negative end of the CV than the middle of the distribution of the Danish. The Early Anglo-Saxons overlap substantially with the Pre-Medieval British on both CVs. In contrast, while the Early Anglo-Saxons overlap with the Danish individuals on CV2, they barely overlap with them on CV3. The Middle Anglo-Saxons have a different relationship with the Pre-Medieval British and Danish individuals. They barely overlap with the Danish on either CV. They overlap markedly more with the Pre-Medieval British on CV2 and even more with them on CV3. The Middle Anglo-Saxons overlap with Early Anglo-Saxons on CV3 but are separated from them on CV2.

The CVA of the male individuals also produced three CVs. CV1 accounted for 44% of the variation, CV2 for 37%, and CV3 for 19%.

Fig 4A shows the values for CV1 and CV2 for the male individuals plotted against each other. The Pre-Medieval British and Danish individuals overlap on both CVs, although to a lesser extent on CV1. On both CVs, the centre of the distribution of Pre-Medieval British individuals is closer to the positive end of the CV than the centre of the distribution of the Danish individuals. The Early Anglo-Saxons overlap substantially with both the Pre-Medieval British and Danish on CV2. The situation is different when we look at CV1. On this CV, the Early Anglo-Saxons overlap with the Danish individuals to a considerable extent, but barely overlap with the Pre-Medieval British ones. The Middle Anglo-Saxons overlap substantially with the Pre-Medieval British, Danish, and Early Anglo-Saxon individuals on CV1. On CV2, they do

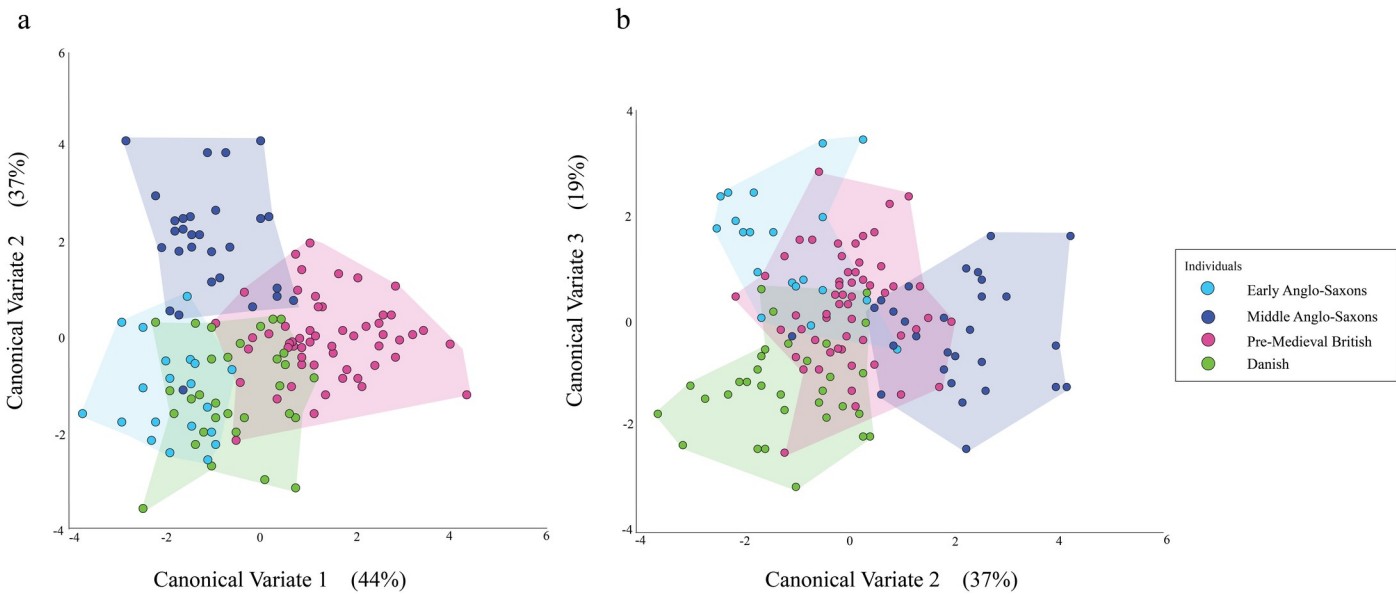

**Fig 4.** Plots of CV values for male individuals when (a) CV1 is plotted against CV2 and (b) CV2 is plotted against CV3.

not overlap at all with the Danish individuals and only slightly with the Pre-Medieval British and Early Anglo-Saxon ones.

The CV2 and CV3 values for the males are plotted against each other in Fig 4B. On CV2, the Danish individuals are positioned towards the negative end of the CV and the Pre-Medieval British ones towards the centre of the CV. The Early Anglo-Saxons overlap substantially with both the Danish and Pre-Medieval British ones on CV2. On CV3, Early Anglo-Saxons overlap to a great extent with the Pre-Medieval British individuals but only partially with the Danish ones. The Middle Anglo-Saxons are distributed in such a way that do not overlap with the Danish individuals on CV2. Instead, they partially overlap with the Pre-Medieval British individuals and extend to the positive end of the CV. On CV3, the Middle Anglo-Saxons overlap substantially with both the Pre-Medieval British and Danish individuals, although the overlap is greater in the case of the Pre-Medieval British ones. The Early Anglo-Saxons and Middle Anglo-Saxons overlap extensively on CV3, but barely overlap on CV2.

The PC reduction procedure retained 10 PCs for the female individuals. Seventy percent of the total shape variance was accounted for by these PCs. The overall MANOVA for the females returned a significant result ($\lambda$ 0.473, F = 2.447, p<0.001, $\eta p^2$ = 0.22). The results of the pairwise MANOVAs are presented in Table 2. Only three of the pair-wise comparisons returned significant results—Pre-Medieval British vs. Danish, Early Anglo-Saxon vs. Middle Anglo-Saxon, and Early Anglo-Saxon vs. Pre-Medieval British.

Twenty-one PCs were retained for the male individuals by the PC reduction procedure. These PCs accounted for 88% of the total shape variance. The overall MANOVA for the males

**Table 2. Results of pairwise MANOVAs.**

| Groups compared | ♀ | ♂ |
|---|---|---|
| Pre-Medieval British *vs* Danish | $\lambda$ 0.613 | $\lambda$ 0.533 |
| | F = 3.3035 | F = 2.756 |
| | p = 0.005* | p<0.001* |
| | $\eta p^2$ = 0.39 | $\eta p^2$ = 0.47 |
| Early Anglo-Saxons *vs* Middle Anglo-Saxon | $\lambda$ 0.553 | $\lambda$ 0.294 |
| | F = 2.343 | F = 3.089 |
| | p = 0.036* | p = 0.003* |
| | $\eta p^2$ = 0.45 | $\eta p^2$ = 0.71 |
| Early Anglo-Saxons *vs* Pre-Medieval British | $\lambda$ 0.610 | $\lambda$ 0.417 |
| | F = 3.903 | F = 3.599 |
| | p<0.001* | p<0.001* |
| | $\eta p^2$ = 0.39 | $\eta p^2$ = 0.58 |
| Early Anglo-Saxons *vs* Danish | $\lambda$ 0.730 | $\lambda$ 0.343 |
| | F = 1.108 | F = 2.741 |
| | p = 0.388 | p = 0.006* |
| | $\eta p^2$ = 0.27 | $\eta p^2$ = 0.66 |
| Middle Anglo-Saxons *vs* Danish | $\lambda$ 0.737 | $\lambda$ 0.525 |
| | F = 1.674 | F = 2.718 |
| | p = 0.115 | p = 0.001* |
| | $\eta p^2$ = 0.26 | $\eta p^2$ = 0.48 |
| Middle Anglo-Saxons *vs* Danish | $\lambda$ 0.517 | $\lambda$ 0.346 |
| | F = 1.495 | F = 3.517 |
| | p = 0.228 | p<0.001* |
| | $\eta p^2$ = 0.48 | $\eta p^2$ = 0.65 |

**Table 3. Summary of the results of the linear discriminant analyses.** The table shows the percentage of Anglo-Saxon individuals attributed to the Pre-Medieval British and Danish groups, and the percentage deemed to be of uncertain ancestry. See the Materials and Methods for details of the criterion used to identify the latter cases.

| | Females | | | Males | | |
|---|---|---|---|---|---|---|
| | Danish | Pre-Medieval British | Unknown | Danish | Pre-Medieval British | Unknown |
| **Early Anglo-Saxons** | 63% | 26% | 11% | 65% | 30% | 5% |
| **Middle Anglo-Saxons** | 39% | 53% | 8% | 31% | 69% | - |

returned a significant result ($\lambda$ 0.268, F = 2.976, p<0.001, $\eta p^2 = 0.36$). All of the pairwise MAN-OVAs returned statistically significant results (Table 2).

Table 3 shows the percentage of the Early Anglo-Saxons and Middle Anglo-Saxons for whom the predicted source population was the Pre-Medieval British group, the percentage for whom the predicted source population was the Danish group, and the percentage deemed unattributable. The individual likelihood frequencies are listed in S3 and S4 Tables in S1 File.

In the female LDA analysis, 63% of the Early Anglo-Saxons were attributed to the Danish group, and 26% to the Pre-Medieval British group. The remaining 11% were deemed to be of uncertain ancestry. Thus, nearly two-thirds of the Early Anglo-Saxons were suggested to be of non-local ancestry, while a quarter were deemed to be of local ancestry. The pattern was different for the Middle Anglo-Saxons. Fifty-four percent of these individuals were attributed to the Pre-Medieval British group; 39% were attributed to the Danish group; and 8% were deemed to be of uncertain ancestry. In other words, just over half of the Middle Anglo-Saxon females were suggested to be of local ancestry and just under half were suggested to be of non-local ancestry.

In the male LDA analysis, 65% of the Early Anglo-Saxons were attributed to the Danish group, and 25% to the Pre-Medieval British group. The remaining 10% were deemed to be of uncertain ancestry. As was the case with the female LDA analysis, the pattern was different for the Middle Anglo-Saxons. Sixty-nine percent of these were attributed to the Pre-Medieval British group, and 31% to the Danish group. As such, the results of the male LDA analysis were similar to those of the female LDA analysis. They also indicated that nearly two-thirds of the Early Anglo-Saxon males were of non-local ancestry while a quarter were of local ancestry. Also like the female data, the male data indicate that the pattern reversed with the Middle Anglo-Saxons, with the majority of them appearing to have been of local ancestry and the minority of non-local ancestry.

## Discussion

In the study reported here, we used 3D shape data from the basicranial portion of the skull to infer the ancestry of a large sample of Anglo-Saxon skeletons from southern Britain and therefore shed light on the scale of the migration from continental northwest Europe that occurred between the mid 5th and early 7th centuries CE. Analyses that focused on Early Anglo-Saxon skeletons indicated that between two-thirds and three-quarters of them were likely of continental northwest European ancestry, while between a quarter and one-third were of local ancestry. Analyses that focused on Middle Anglo-Saxon skeletons returned substantially different results. They indicated that 50–70% of the Anglo-Saxon individuals were likely of local ancestry, while 30–50% of them were likely of continental northwest European ancestry.

To ensure that our findings were robust, we ran two supplementary analyses. In the first, we checked that our decision to keep the females and males separate did not skew our results. We created a mixed-sex dataset and then subjected it to LDA in the manner described earlier. Of the 47 Early Anglo-Saxons, 72% were assigned to the Danish group, 23% to the Pre-Medieval British group, and 4% were deemed unattributable. Of the 42 Middle Anglo-Saxons, 52%

were assigned to the Pre-Medieval British group, 45% to the Danish group, and 2% were deemed unattributable. These results are similar to those obtained in the sex-specific LDAs, which indicates that our decision to keep the females and males separate did not skew our results.

In the second supplementary analysis, we estimated the contributions of the two potential source groups to the ancestry of the Early and Middle Anglo-Saxons in a different manner. In the main analyses, we used the LDA probabilities to assign each Anglo-Saxon individual to a single potential source group and then calculated the percentage of Anglo-Saxon individuals assigned to each of the two potential source groups. However, it is also possible to treat the LDA percentages as estimates of the contribution of the two potential source groups to each Anglo-Saxon individual's ancestry and then use the average percentages as the estimates of the contributions of the two potential source groups to the ancestry of Early and Middle Anglo-Saxons. The main difference between these approaches is that the former assumes that the Anglo-Saxons are either locals or non-locals, whereas the latter allows for the possibility that some of the Anglo-Saxons were the products of intermarriage between locals and non-locals.

When the Early Anglo-Saxon females were subjected to the second approach, 69% of their ancestry was estimated to derive from Denmark and 31% from Pre-Medieval Britain. This changed with the Middle Anglo-Saxon females, with 52% of their ancestry being estimated to derive from Pre-Medieval Britain and 48% from Denmark (S3 Table in S1 File). When the same was done for the Early Anglo-Saxon males, 70% of their ancestry was estimated to derive from Denmark and 30% from Pre-Medieval Britain. As with the females, the pattern changed with the Middle Anglo-Saxon males. Sixty-three percent of their ancestry was estimated to come from Pre-Medieval Britain and 37% from Denmark (S4 Table in S1 File). Thus, estimating the contributions of the two potential source groups to the ancestry of the Early and Middle Anglo-Saxons in a different way did not change our findings.

As explained previously, three other lines of evidence have been used to investigate the relative number of individuals of local ancestry and individuals of continental northwest European ancestry among the Anglo-Saxons: historical texts, isotopes, and DNA. To reiterate, the relevant historical texts, the *Historia Ecclesiastica* and the *Anglo-Saxon Chronicles*, describe a wholesale replacement of the native population by a large number of Germanic-speaking people from continental northwest Europe. A plausible explanation for the discrepancy between our results and the historical texts is that the latter are simply inaccurate. As Hamerow [38] and Thomas et al. [39] have pointed out, the *Historia Ecclesiastica* and the *Anglo-Saxon Chronicles* were both written several centuries after the events in question and therefore it is feasible that their authors greatly exaggerated the number of settlers. One possibility is that continental northwest European ancestry was more prestigious than local ancestry and that prompted the majority of later Anglo-Saxons to claim to have continental northwest European ancestors. Another interesting possibility is raised by Reynolds' [1, pp 399] work. She argued that Medieval people believed that humans were "divided into 'peoples' (*gentes*, *nationes*, *populi*) of common biological descent and culture who normally and naturally formed separate political units". It is not hard to see how this belief could have led to the creation of a founding myth in which the Anglo-Saxons were solely the descendants of Germanic-speaking settlers.

A number of isotopic studies have focused on Anglo-Saxon skeletons. For example, Budd et al. [13] analysed oxygen isotope ratios in 32 individuals from the cemetery of West Heslerton, North Yorkshire, and concluded approximately half of the samples analysed were from other areas of Britain, while four females appeared to originate from Scandinavia or Baltic Europe. Subsequently, Hughes et al. [14] extracted strontium and oxygen isotopes from the remains of 19 Anglo-Saxon individuals from Berinsfield, Oxfordshire. Four of 19 individuals were found to have isotope values indicating they did not spend their childhood in the area.

One of these individuals was deemed to be from mainland Europe, while the provenance of the other three was deemed uncertain and the most the authors could say was that they were not local. In line with their previous results, Hughes et al. [14] identified seven individuals of 19 from the Early Anglo-Saxon cemetery in Eastbourne, Sussex, with non-local strontium isotope values. Four of these seven were interpreted to be from mainland Europe. Thus, the isotope studies suggest that only a small number of Anglo-Saxons were from mainland Europe.

There would seem to be several potential explanations for the discrepancy between the results of the isotope studies and our study's findings. To begin with, it is possible that the discrepancy is more apparent than real. Analyses of oxygen and strontium isotopes extracted from teeth shed light on where an individual grew up rather than their ancestry. So, it is feasible that some, if not all, of the individuals who were deemed to be local had parents raised on the Continent but were themselves raised in southern Britain, i.e. they were second-generation settlers. If this were the case, then the results reported here and those of the isotope studies are not inconsistent. A second possibility is that there was regional variation in the number of Germanic-speaking settlers. None of the Anglo-Saxon skeletons we analysed are among those sampled in the isotope studies. So, it is possible that both sets of results are correct and that the number of Germanic-speaking settlers was simply higher in the areas of southern Britain represented in the isotope studies than the area represented in the present study. A third possibility is that the isotope studies underestimated the number of individuals from the Continent among the Anglo-Saxons. All three studies identified additional non-local individuals in their samples. Some of these individuals were hypothesised to be from elsewhere in the British Isles, while others were deemed to be of uncertain provenance. However, there is a problem with these assessments. They depend on good data on landscape $^{87}Sr/^{86}Sr$ isotope ratio variability in the regions of interest. As Bataille et al. [40] demonstrate, while the British Isles and the homeland of some of the incoming people have been well sampled for $^{87}Sr/^{86}Sr$ isotope ratios, the data for the other areas of interest in continental Europe are limited and patchy. This raises the possibility that some of the individuals who were identified as having been from elsewhere in the British Isles could in fact have been from continental northwest Europe. The same holds for the individuals that the authors were unable to provenance.

The earliest of the DNA studies to have focused on the origins of the Anglo-Saxons estimated that 50–100% of males in England have mainland European ancestry [3]. Subsequent studies have estimated the percentage of mainland European ancestry in England to be between 10% and 73% [4, 16, 17]. Thus, in contrast to the situation with the historical texts and perhaps with the isotope data, our results are not inconsistent with the available DNA evidence regarding the ancestry of the Anglo-Saxons. Our estimate of between two-thirds and three-quarters of the Early Anglo-Saxons having been of continental northwest Europe ancestry falls comfortably within the range of estimates obtained in the DNA studies. So does our estimate of between a third and a half of Middle Anglo-Saxons having been of continental northwest Europe ancestry. The compatibility between the findings of the DNA studies and the results of the present study supports the use of 3D basicranial shape as an indicator of ancestry in archaeological human skeletons [3, 4, 16, 17].

It is intriguing that our results indicate that the number of Anglo-Saxons of continental northwest European ancestry decreased from about two-thirds in the Early Anglo-Saxon Period to around one-third in the Middle Anglo-Saxon Period. To the best of our knowledge, a marked change in relative numbers of locals and non-locals between these periods has not been identified before. The historical texts do not mention one, nor do the isotope and DNA studies. There are a number of potential explanations for such a change. To begin with, it could mean that there was an increase in the number of local people adopting the Anglo-Saxon identity from the third to the seventh century. Alternatively, it could mean that mass

migration of peoples from mainland Europe had ceased and there was an increase in intermarriage between people of continental northwest European ancestry and those of local ancestry. A third possibility is that there was still migration from mainland Europe after 660 CE but at a much lower level than in the preceding period. Lastly, it is possible that the increase in the percentage of individuals of local ancestry was a consequence of those with local ancestry out-reproducing those of continental northwest European ancestry. Determining which, if any, of these hypotheses is correct will require further research.

There are two other obvious possibilities for future research. One is to repeat the study with a more comprehensive sample. In particular, it would also be useful to include individuals from regions of Britain not represented in the present study (see Fig 1). This would allow the hypothesis that there was regional variation in the relative number of Anglo-Saxons of continental northwest European and local ancestry to be evaluated. A more comprehensive sample would also allow the possibility that there were sex differences in migration and acculturation to be investigated. In the present study, the results for males and females were similar. However, other studies have suggested that the majority of incomers were male [3]. Needless to say, it would be good to resolve this discrepancy.

The other obvious next step is to repeat the study with data from one or more other regions of the skull. We focused on the basicranium because previous analyses have found that 3D shape correlates significantly with among-population genetic distance in humans, but studies in palaeoanthropology have suggested that there are no major differences among the cranial regions with regard to inferring ancestry [41]. With this in mind, it would be useful to apply 3D geometric morphometric techniques to the faces, cranial vaults, and/or mandibles of a large sample of suitably dated individuals. Doing so could allow one of the major shortcomings of the present study to be overcome—namely, the inability to include individuals from Germany and the Netherlands due to poor preservation of the cranial base.

## Conclusions

The study reported here focused on an important unresolved issue in British history—the relative number of individuals of local ancestry and individuals of continental northwest European ancestry among the Anglo-Saxons. Existing lines of evidence do not agree on this issue. Historical texts indicate that the vast majority of Anglo-Saxons were of continental northwest European ancestry, while stable isotope analyses suggest that there were in fact only a few individuals from mainland Europe among the Anglo-Saxons. Unfortunately, analyses of DNA have not clarified the situation. The results that have been reported to date are highly variable, with estimates of the percentage of continental northwest European ancestry in England ranging between 10% and 100%, depending on the study.

We sought to reduce the uncertainty by comparing the basicrania of individuals from five Anglo-Saxon cemeteries to the basicrania of individuals from sites in the British Isles that predate the Medieval period, and sites in Denmark that date to the Iron Age. Denmark was the homeland of two of the Germanic-speaking groups whose members settled in southern Britain between the mid 5th and early 7th centuries CE and contributed to the formation of the Anglo-Saxon ethnic group. Thus, the comparison allowed us to estimate the percentage of Anglo-Saxon individuals who were of local ancestry and the percentage who were of continental northwest European ancestry.

Analyses that focused on skeletons from the Early Anglo-Saxon Period (410 CE-660 CE) indicated that between two-thirds and three-quarters of the Anglo-Saxon individuals were likely of continental northwest Europe ancestry, while between a quarter and one-third of them were people of local ancestry. Analyses that focused on individuals from the Middle

Anglo-Saxon Period (660 CE-889 CE) returned substantially different results. They indicated that 50–70% of the Anglo-Saxon individuals were likely of local ancestry, while 30–50% of them were likely of continental northwest European ancestry.

Our study suggests, therefore, that the Anglo-Saxon ethnic group comprised many individuals of continental northwest European ancestry but also many of local ancestry. Additionally, our study suggests that the relative number of individuals of continental northwest European ancestry and individuals of local ancestry changed through time. In the Early Anglo-Saxon Period, individuals of continental northwest European ancestry heavily outnumbered individuals of local ancestry, but by the Middle Anglo-Saxon Period, those of local ancestry outnumbered those of continental northwest European ancestry. Intriguingly, it appears that ancestry in Early Medieval Britain was similar to what it is today—mixed and mutable.

## Supporting information

**S1 File.**
(DOCX)

**S2 File.**
(TXT)

## Acknowledgments

We offer special thanks to Carly Ameen and Allowen Evin for their assistance with the analyses. We also thank Neil Price and Alex Woolf for their advice regarding the manuscript. The following institutions kindly provided access to the collections used in the study: University of Copenhagen, Copenhagen, Denmark; the National Museum of Natural History, London, UK; Tarbat Discovery Centre, Portmahomack, UK; Royal College of Surgeons, London, UK; Kent Archaeology Trust, Dover, UK; Canterbury Archaeological Trust, UK; National Museum of Scotland, Edinburgh, UK; the Duckworth Laboratory at the University of Cambridge, Cambridge, UK. Lastly, we thank the editor and five anonymous reviewers for their assistance with the manuscript. We appreciate their time and trouble.

## Author Contributions

**Conceptualization:** Kimberly A. Plomp, Keith Dobney, Mark Collard.

**Data curation:** Kimberly A. Plomp.

**Formal analysis:** Kimberly A. Plomp.

**Funding acquisition:** Kimberly A. Plomp, Keith Dobney, Mark Collard.

**Investigation:** Kimberly A. Plomp, Keith Dobney.

**Methodology:** Kimberly A. Plomp.

**Project administration:** Kimberly A. Plomp.

**Supervision:** Keith Dobney, Mark Collard.

**Validation:** Kimberly A. Plomp, Keith Dobney, Mark Collard.

**Visualization:** Kimberly A. Plomp, Keith Dobney, Mark Collard.

**Writing – original draft:** Kimberly A. Plomp, Keith Dobney, Mark Collard.

**Writing – review & editing:** Kimberly A. Plomp, Keith Dobney, Mark Collard.

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
