## [Decision Letter · Decision Letter 0]

24 Mar 2021

PONE-D-21-04061

A 3D basicranial shape-based assessment of local and continental northwest European ancestry among 5th to 9th century CE Anglo-Saxons

PLOS ONE

Dear Dr. Plomp,

Thank you for submitting your manuscript to PLOS ONE. After careful consideration, we feel that it has merit but does not fully meet PLOS ONE’s publication criteria as it currently stands. Therefore, we invite you to submit a revised version of the manuscript that addresses the points raised during the review process.

All reviewers found this work to be of great interest. However, there was a consistent concern over statistical approach and interpretation of those results. There are other minor issues to be addressed as well.

We look forward to receiving your revised manuscript.

Kind regards,

JJ Cray Jr., Ph.D.

Academic Editor

PLOS ONE

Journal Requirements:

2. In your manuscript, please provide additional information regarding the specimens used in your study. Ensure that you have reported specimen numbers and complete repository information, including museum name and geographic location.

For more information on PLOS ONE's requirements for paleontology and archaeology research, see https://journals.plos.org/plosone/s/submission-guidelines#loc-paleontology-and-archaeology-research.

3. Please include a copy of Table 1 which you refer to in your methods section.

Reviewers' comments:

Reviewer's Responses to Questions

**Comments to the Author**

1. Is the manuscript technically sound, and do the data support the conclusions?

Reviewer #1: Partly

Reviewer #2: Partly

Reviewer #3: Yes

Reviewer #4: Yes

Reviewer #5: Partly

2. Has the statistical analysis been performed appropriately and rigorously? 

Reviewer #1: No

Reviewer #2: Yes

Reviewer #3: Yes

Reviewer #4: Yes

Reviewer #5: Yes

3. Have the authors made all data underlying the findings in their manuscript fully available?

Reviewer #1: Yes

Reviewer #2: Yes

Reviewer #3: Yes

Reviewer #4: Yes

Reviewer #5: Yes

4. Is the manuscript presented in an intelligible fashion and written in standard English?

Reviewer #1: Yes

Reviewer #2: Yes

Reviewer #3: Yes

Reviewer #4: Yes

Reviewer #5: Yes

5. Review Comments to the Author

Reviewer #1: The manuscript by Plomp and colleagues explores the morphological affinities of Anglo-Saxon skeletal collections, with the goal of contributing to the discussion about the impact of continental populations in the formation of Anglo-Saxon biological diversity. The question presented is of interest and relevance, given that it has been discussed by many authors from different disciplines. As such it could be of relevance for an inter-disciplinary audience and therefore could be suited for PLoS One.

However, there are a series of aspects of the methods adopted that should be clarified or revised before I can recommend the manuscript for publication. I list below my questions and comments, which I hope will help the authors to revise the manuscript.

1. My main concern with the current version of the manuscript relates to the use and interpretation of the results from the classification of the individuals based on the LDA, as I believe the authors are not properly taking into account the inherent limitations of using modern human morphology to infer ancestry.

a. The first aspect of these limitations to consider is that, even for the anatomical regions that best discriminate modern human populations, there is considerable overlap among them. Because of that, it would be important to either demonstrate that the source populations can be successfully and reliably discriminated and/or take into account the uncertainty in the classification per se. The authors demonstrate through MANOVAs that the source populations are different, but this is based on PCs and not on the CVAs. So, my first suggestion would be to test how often the Danish individuals classify as pre-Medieval English, and vice-versa. If only a few of the individuals from each source population classify incorrectly, this supports the notion that these populations have morphological differences that allow for reliable discrimination. If that is not case, however (which would be expected for modern human populations), the results of the classification of the earlier Anglo-Saxons must be presented more cautiously (see my next point).

b. The second approach to consider is to incorporate better the uncertainty of the classification results. Because the classification is usually done based on the smallest distance to the centroids of the reference populations, in situations where there is a lot of overlap in the variance of the series, individuals who belong to a population can be frequently misclassified. It is not clear in M&M (see my next point) exactly how the classification was done, but it is important that if it is based on posterior probabilities or typicalities that the authors consider the probabilities of belonging to each source population. Having a higher probability of belonging to group A does not necessarily reject the hypothesis that it could belong to group B (which is ultimately the hypothesis being tested in this study). If the source populations are not that distinct, which is a possibility here, it is particularly important to be more explicit about how the authors are considering false-positive affiliations.

2. The Material and Methods section should be more explicit about how the analyses were conducted. In particular for the classification, it is hard to understand how it was done. For example, it is not clear to me what the authors mean by: “We repeated this process 100 times for each Anglo-Saxon individual and then calculated the average attribution percentages.” Why was the analysis repeated a 100 times? Was the source data permutated somehow? Why not use more traditional classification analyses and cross-validation through posterior probabilities? I am not saying the authors should change the method, just that I do not understand what the method is. Therefore, more justification is required here.

3. Similarly, I do not understand why the authors chose 55% as the cut value of membership to each population. It seems unreasonable that this is the cut-value, especially since the source populations may be very similar to start with. This value must be justified better, but I strongly recommend the authors adopt a more conservative value here to avoid false positives.

4. While it is a nice visual exploratory analysis, I do not see how the CVA plots are informing the conclusions of the manuscript in any meaningful way. I suggest the authors try to integrate the patterns observed from the CVA plots more explicit in their interpretations.

5. It would be helpful if the MANOVAS results between pairs of series were presented in a table, to facilitate the visualization of the results.

6. Finally, I think the discussion is also not giving sufficient attention to the limitations of the morphological analyses. I agree with the authors that isotopic analyses are limited and that their results can be biased. But it must also be considered that the reason of the discrepancy may be due to the limitations of morphological analyses. At least, it would be important for the authors to argue explicitly why we should consider their results more reliable than other sources of information. Once again, I don’t say this because I think morphological data is not reliable, which it is, but it is important to address its limitations more explicitly, especially when writing for a journal that is targeting a broader inter-disciplinary audience.

Reviewer #2: This manuscript provides important data on the population history of Anglo-Saxon British populations and demonstrates that Early Medieval Britain was heterogeneous in terms of genetic ancestry. The authors use well accepted geometric morphometrics and linear discriminant function analyses (LDA) to analyze individuals from Anglo-Saxon, pre-Medieval, and Iron Age/Viking sites. In so doing, they demonstrate that Early Medieval populations were comprised of both local and non-local (from mainland Europe) groups, which both contradicts and supports previous research that involves historical, stable isotope, and genetic data. Additionally, the population structures change through time. Overall, the manuscript is well written and distills the information down in a very understandable way while contributing to an important dialog around the peopling of Britain.

The comments and suggestions are primarily minor in nature and are documented in the attached PDF. I'll briefly highlight the main ones here:

The authors indicate that their sample size is 237; however, I came up with 236.

If possible, I'd like to see some more discussion regarding the sex estimations of the individuals in their sample, since this is a key foundation of their study. The authors indicate that they based the sex estimations on the pelvis and skull. It does not look like there were any indeterminate/ambiguous skeletons, which is usually rare. Were all individuals placed into female or male categories based on the pelvis and skull, or was the skull or pelvis used? What if the skull and pelvis "disagreed"?

A paragraph in the Discussion that addresses the results when pooling females and males reads, to me, more like it belongs in the Results. I suggest it be moved to Results and a brief statement added in the Methods.

I do not see Table 1 that is referenced in the narrative (only Table 2 and Table S1).

I am curious whether the LDAs were cross-validated. Cross-validation can ensure that the individual under analysis was not included in the method development and is a more accurate reflection of an unknown individual.

The last sentence of the Abstract and Conclusion state that "identity in Early Medieval Britain was similar to what it is today--inclusive and dynamic." While this may be true, this study does not examine social identity of past or present peoples (nor the bioarchaeological literature on identity). Ultimately, we do not know how people may have identified (or identified others) and whether such social identities map onto the genetic/ancestral data. I concur that the population was heterogeneous and dynamic in terms of ancestry, but not necessarily in identity. Also, heterogeneity doesn't necessarily equate to "inclusivity." It may be that there were invasions or forced migrations, which arguable isn't inclusion (making space for others). It may also be argued that modern Britain, like anywhere, isn't inclusive. I suggest limiting this conclusion to the heterogeneous and dynamic nature of the ancestries/genetics and avoiding reference to identity altogether.

I look forward to seeing this revised and published!

Reviewer #3: This paper provides a novel approach to address longstanding questions regarding human migration patterns. I believe it should be published with minor revisions, as discussed below:

In the Materials and Methods section, it is stated that the sample is summarized in Table 1, but there is no Table 1 included in the text. This table should be added to clarify the demographics of the sample.

It would be helpful to include date ranges for the Danish sites, as is done with the other sites.

Each anatomical landmark that was collected should be named and/or described, either within the text, in a new table, or as part of Figure 2.

In Figures 3 and 4, it would be helpful to include the percentage of variance explained along each axis in each of the axis labels.

Would it be possible to describe the variance found along the different CV axes in anatomical terms? What anatomical patterns are described along each axis?

The second paragraph of the Discussion section is repetitive, using the phrase “our decision to keep the female and male specimens separate did not skew our results” twice. This should be reworded to avoid this repetition.

I appreciate the authors’ thoughtful comparison of their results to previously published historical, isotope, and genetic evidence.

In the seventh paragraph of the Discussion section, there is an inappropriate use of a semicolon, “The historical texts do not mention one; nor do the isotope and DNA studies.” This semicolon should be replaced with a comma.

It is assumed in the study that variations between the Early and Middle Anglo-Saxon samples are caused by differing patterns of ancestry, but is it possible that there are other factors influencing these differences? Does basicranial morphology change over time in the Danish or pre-Medieval samples?

In the Supplementary Document, there are two tables both labeled as Supplementary Table 1. Change the title of one of these tables to Supplementary Table 2. Also, the titles of these tables should be changed to “female” and “male” rather than “females” and “males” since the term “each” is used.

Reviewer #4: 1. Summary of the research and your overall impression

The authors have designed a novel study to identify the ancestral origins of Anglo-Saxon peoples in Medieval Europe and trace the path of migration across time. Previous research has used historical, archeological, or genetic evidence to track the migration of mainland European peoples to the British Isles. However, these data are incongruent and tell conflicting stories. The current study importantly adds a new line of evidence to paint a clearer picture of this very complex human phenomenon to see how mainland European and local populations interacted with one another. The authors employed cutting-edge technology to acquire biological data from the skeletons of historic archeological populations. Then, geometric morphometric techniques were used to analyze the data. Specifically, three-dimensional coordinate landmark data were abstracted from digital photogrammetric reconstructions of a relatively large sample of skulls. Using statistical methods borrowed from paleontology, the basicranial shape of three major groups separated by time and space were compared to identify morphological similarities; thus, a proxy for migration and ancestry is given. The authors found that Early Anglo-Saxons were mostly derived from continental Europe populations whereas Middle Anglo-Saxons were mostly derived from local populations. The proposals for future research give readers possibilities to explore this topic further and enhance the accuracy of the data.

The manuscript is easy to follow and sensibly organized. The introduction proceeds from general to specific ideas and flows into the methodology. The results are clearly presented and are supported by the data. The conclusions can be traced back through the results and answer the original question. The primary strength is the relatively large sample size of 237 individuals. Studies involving skeletal collections are typically abhorrently small due to access or preservation issues. This sample size allows for the authors to make more generalizable inferences about ancestral origins. Additionally, the use of landmark morphometric techniques to collect and analyze data, in conjunction with contextual evidence, is relatively novel for bioarcheological studies such as this. The authors do well at plainly explaining their methodology without getting caught up with the theoretical and mathematical underpinnings. However, there are a couple details that were left out of the methodology to make the study reproducible.

I recommend that this study be accepted for publication after minor revisions.

2. Evidence and examples

Major Issues

1. No major issues noted.

Minor Issues

1. Table 1 referenced in the first line of the Material and Methods section is absent in the manuscript. If a table is not provided showing sub-sample sizes, including the number of males and females for each group, along with totals, a description of the sample breakdown should be given in the text.

2. A list of the 34 landmarks used is not given. Page 6, last paragraph refers to landmarks used by Havarti and Weaver (2006) and a figure is referenced in the current study. Once could infer the names of said landmarks, but without labels the reader does not know precisely which landmarks were used. Thus, the study could not be repeated. Include a table listing these landmarks and label the points in Figure 2 with numbers cross-referenced to the new table. A brief summary of how many landmarks are midline and how many are bilateral would be a helpful, as commonly seen in similar studies of bilaterally symmetric structures.

3. The observer error sub-study described on page 7 first full paragraph does not indicate whether there was a single or multiple observers. I inferred a single observer and was able to confirm this in the “Authors’ Contributions” section. A mention of a single observer in the text would mitigate expectations for inter-observer error test results.

4. The computer program MorphoJ is incorrectly referenced. Dr. Klingenberg’s lab website (https://morphometrics.uk/MorphoJ_page.html) indicates the preferred citation to be Klingenberg, C. P. 2011. MorphoJ: an integrated software package for geometric morphometrics. Molecular Ecology Resources 11: 353-357.

5. The authors use an uncommon or possibly incorrect symbol for Wilk’s lambda used in reporting MANOVA test results on page 12. Wilk’s lambda is usually abbreviated with the upper-case lambda symbol (Λ) or less commonly a lower-case lambda (λ). The barred lambda given in the manuscript may be a symbol formatting error.

6. The presentation of results is very clear and easy to follow. The authors do not include the effect size for MANOVA results on page 7. Including the partial eta-squared value will help assess the magnitude of the difference in basicranial shape between the groups. Following Cohen J (1992). A power primer. Psych Bull, 112, 155-159, once can interpret partial eta-squared effect sizes as 0.01 = small, 0.06 = medium, and 0.14 = large. A p-value can only tell you if there is a difference or not, but the effect size indicates how large or small that difference is. This may help to tease out subtle differences between groups as statistically significant, but practically insignificant.

7. The two other possibilities for future results on page 19 are well considered. For the first suggestion, the authors should also contemplate including an outgroup in future work to act as a control. Discriminant analysis creates a function that maximizes the differences between groups, no matter how small or practically insignificant they may be. The incorporation of another population removed by geography and/or time can help show that the methodology used is working as it should.

8. Figures 3 and 4 are well laid out but have axis titles and labels that are difficult to read. The numbers are generally referenced in the text, but the figures are difficult to interpret. Also, the captions for Figures 3 and 4 do not give titles for a or b. ‘a’ is CV1 versus CV2 and ‘b’ is CV2 versus CV3.

9. The authors interchangeably use the terms “individuals” in place of “specimens” when referring to once living human beings. The connotation of “individual” maintains the humanity of these persons while they provide us with information about our past. Using these terms will help overcome anthropology’s contentious history of digging up the dead without permission.

Reviewer #5: The paper investigates craniometric variation of pre-medieval and medieval populations from the British Isles and Denmark in an effort to further reconstruct population history of the "Anglo-Saxons". It focuses on the basicranium as proxy for relatedness, from which it infers ancestry. Based on multivariate statistics (Manova, LDA) the authors conclude that the continental ancestry of Anglo-Saxons decreased from the early to a later phase.

The paper is generally well written, the methods are those that are commonly used when dealing with morphometric variation, the samples are not huge, but probably sufficient to address the general question the study is aimed at. Some methodological aspects require clarification and revision prior to publication.

Major

The first CVA computed on the Procrustes residuals of males and females separately probably does not yield reliable results. By my account the number of variables (p*d=102) is 3times larger than any of the groups, which is not in line with general recommendations (e.g. Bookstein & Mitteroecker, 2011).

The summary of the Manova is confusing: it reports significant differences for pairwise-comparisons and a different value for Wilks' lambda for each of these comparisons. Normally, one would expect one Wilks' lambda for the female Manova and one for the male, so it is not clear what is being reported here and how p-values for pairwise comparisons were obtained.

The key results are obtained by LDA, to classify the Anglo-Saxon samples as either pre-Medieval British or as Danish. The authors find a shift from high to low Danish classification rates between early and Middle Anglo-Saxons. I am a bit skeptical about the female result (only 13 Middle_Anglo-Saxons): It is consistent with the larger male sample and suggests that continental ancestry shifted similarly in males and females. The authors repeated the LDA on a pooled sample to ascertain that the results from the sex-specific analyses are not biased. However, the pooled middle Anglo-Saxon sample remains predominantly male, so some uncertainty remains, as does the possibility of differential ancestry in males and females. Considering that some genetic data suggest high male continental ancestry, it would be worth pursuing this question, though I reckon this would require additional samples from Middle Anglo-Saxon females.

Minor

Table 1 lists only the Anglo-Saxon samples, omitting info for sex of the pre-Medieval and Danish samples. It provides results of the LDA individually, which is ok for a supplementary document. I would suggest including a more synthetic version (e.g. sample size by sex/chronology/geography) in the text, since this information is important for assessing the scope of the study.

6. PLOS authors have the option to publish the peer review history of their article (what does this mean?). If published, this will include your full peer review and any attached files.

Reviewer #1: No

Reviewer #2: No

Reviewer #3: No

Reviewer #4: No

Reviewer #5: No

---

## [Author Response · Author response to Decision Letter 0]

8 Apr 2021

Responses to reviewers’ comments on Plomp et al.’s ‘A 3D basicranial shape-based assessment of local and continental northwest European ancestry among 5th to 9th century CE Anglo-Saxons’

Our responses to reviewers’ comments are below. Reviewer comments are in black text and our responses are in blue.

Editorial comments

“In your manuscript, please provide additional information regarding the specimens used in your study. Ensure that you have reported specimen numbers and complete repository information, including museum name and geographic location.”

Done.

“If permits were required, please ensure that you have provided details for all permits that were obtained, including the full name of the issuing authority, and add the following statement: 'All necessary permits were obtained for the described study, which complied with all relevant regulations.'

If no permits were required, please include the following statement: ‘No permits were required for the described study, which complied with all relevant regulations.’ For more information on PLOS ONE's requirements for paleontology and archaeology research, see https://journals.plos.org/plosone/s/submission-guidelines#loc-paleontology-and-archaeology-research.”

Done – Line 190.

“Please include a copy of Table 1 which you refer to in your methods section.”

Done.

“Your ethics statement should only appear in the Methods section of your manuscript. If your ethics statement is written in any section besides the Methods, please move it to the Methods section and delete it from any other section. Please ensure that your ethics statement is included in your manuscript, as the ethics statement entered into the online submission form will not be published alongside your manuscript.”

Done. – Lines 187-192.

Reviewer #1

“My main concern with the current version of the manuscript relates to the use and interpretation of the results from the classification of the individuals based on the LDA, as I believe the authors are not properly taking into account the inherent limitations of using modern human morphology to infer ancestry.

a. The first aspect of these limitations to consider is that, even for the anatomical regions that best discriminate modern human populations, there is considerable overlap among them. Because of that, it would be important to either demonstrate that the source populations can be successfully and reliably discriminated and/or take into account the uncertainty in the classification per se. The authors demonstrate through MANOVAs that the source populations are different, but this is based on PCs and not on the CVAs. So, my first suggestion would be to test how often the Danish individuals classify as pre-Medieval English, and vice-versa. If only a few of the individuals from each source population classify incorrectly, this supports the notion that these populations have morphological differences that allow for reliable discrimination. If that is not case, however (which would be expected for modern human populations), the results of the classification of the earlier Anglo-Saxons must be presented more cautiously (see my next point).

We appreciate the reviewer’s concern but running the MANOVAs and LDAs with the CVA scores would not have been the right course of action, in our view. The reason for this is that CVA maximizes between-group variation relative to within-group variation on the basis of pre-specified groups. PCA does not do this. Thus, using the CVAs in MANOVAs and LDAs would have been more permissive / less conservative than using the PC scores. That said, when running a CVA it is possible to run permutations on group classification to obtain pair-wise p-values indicating the statistical differences between groups. We have done this, and the Pre-Medieval British and Danish sub-samples are significantly different for both males and females, which supports the MANOVAs reported in the manuscript.

Running LDAs on a random sample of Pre-Medieval British and Danish individuals as ‘unknowns’ would not shed light on the reliability of our results because the size of the two sub-samples would be reduced relative to the equivalent samples used in the Anglo-Saxon-focused analyses.

b. The second approach to consider is to incorporate better the uncertainty of the classification results. Because the classification is usually done based on the smallest distance to the centroids of the reference populations, in situations where there is a lot of overlap in the variance of the series, individuals who belong to a population can be frequently misclassified. It is not clear in M&M (see my next point) exactly how the classification was done, but it is important that if it is based on posterior probabilities or typicalities that the authors consider the probabilities of belonging to each source population. Having a higher probability of belonging to group A does not necessarily reject the hypothesis that it could belong to group B (which is ultimately the hypothesis being tested in this study). If the source populations are not that distinct, which is a possibility here, it is particularly important to be more explicit about how the authors are considering false-positive affiliations.”

The LDA results were cross-validated, which should limit the number of false-positive affiliations present in the results. This information has now been added to the Methods section (line 173). In addition, we have now tried to estimate British and Danish ancestry in the Anglo-Saxon samples in an additional way. To reiterate, in the original analysis we assigned individual Anglo-Saxon specimens to the British group or the Danish group and then calculated the percentage that belonged to each group. In the new analysis, we treated the LDA percentages as estimates of the contribution of the two potential source groups to each Anglo-Saxon individual’s ancestry and then used the average percentages as the estimates of the contributions of the two potential source groups to the ancestry of Early and Middle Anglo-Saxons. This analysis sidesteps the issue highlighted by the reviewer. Significantly, the results of the two analyses are consistent. We have added a description of the new analysis to the Discussion.

“2. The Material and Methods section should be more explicit about how the analyses were conducted. In particular for the classification, it is hard to understand how it was done. For example, it is not clear to me what the authors mean by: “We repeated this process 100 times for each Anglo-Saxon individual and then calculated the average attribution percentages.” Why was the analysis repeated a 100 times? Was the source data permutated somehow? Why not use more traditional classification analyses and cross-validation through posterior probabilities? I am not saying the authors should change the method, just that I do not understand what the method is. Therefore, more justification is required here.”

The LDA results were cross-validated during the LDA using the pldam source code in R. The sentence brought up here was erroneously left in from an earlier draft and has now been removed. We have also added in the detail that the LDAs were cross-validated.

“Similarly, I do not understand why the authors chose 55% as the cut value of membership to each population. It seems unreasonable that this is the cut-value, especially since the source populations may be very similar to start with. This value must be justified better, but I strongly recommend the authors adopt a more conservative value here to avoid false positives.”

The standard cut-off when there are two potential origin populations is 50% (e.g. Evin et al. 2015; Boedeker and Kearns 2019), so 55% is conservative. Based on the LDA results presented in Supplementary Tables 3 and 4, we could change the cut value to 60% if the reviewer and editor would like, but this would only change the status of a single Early Anglo-Saxon male, BoH24, which had a 57% likelihood of being a member of the Pre-Medieval British group. Considering this, we argue that increasing the cut value for the LDAs would not alter our results in a meaningful way. We have added this to the appropriate paragraph in the Methods section – “Since there are two potential source groups, the standard average percentage used to attribute an individual to one of the source groups is 50% [36, 37]. We opted for a more conservative affiliation value, and so an individual was deemed to be attributed to a given source group if the average percentage for that group was ≥55%” Lines 180-184.

“While it is a nice visual exploratory analysis, I do not see how the CVA plots are informing the conclusions of the manuscript in any meaningful way. I suggest the authors try to integrate the patterns observed from the CVA plots more explicit in their interpretations.”

We disagree with this comment. The variance depicted in the CVA scatter-plots (Figures 3 and 4) are consistent with the LDA results because they indicate that Early Anglo-Saxons tend to group more closely with the Danish individuals, while the Middle Anglo-Saxons tend to group more closely with Pre-Medieval British individuals.

“It would be helpful if the MANOVAS results between pairs of series were presented in a table, to facilitate the visualization of the results.”

The MANOVA results have now been summarised in a table (new Table 2)—see line 24.

“Finally, I think the discussion is also not giving sufficient attention to the limitations of the morphological analyses. I agree with the authors that isotopic analyses are limited and that their results can be biased. But it must also be considered that the reason of the discrepancy may be due to the limitations of morphological analyses. At least, it would be important for the authors to argue explicitly why we should consider their results more reliable than other sources of information. Once again, I don’t say this because I think morphological data is not reliable, which it is, but it is important to address its limitations more explicitly, especially when writing for a journal that is targeting a broader inter-disciplinary audience.”

First, as we explain in the Discussion, there is not necessarily a discrepancy between the results of the isotope studies and our results. If some of the Anglo-Saxon specimens for which isotope results are available are assumed to be second generation immigrants, the discrepancy disappears. Second, we believe that the alignment of our results with the estimates of ancestry provided by the DNA analyses indicates that our results are more reliable than . To emphasize this point, we have added a sentence at the end of the DNA paragraph – “The compatibility between the findings of the DNA studies and the results of the present study supports the use of 3D basicranial shape as an indicator of ancestry in archaeological human skeletons.”

*****

Reviewer #2

“The authors indicate that their sample size is 237; however, I came up with 236.”

Thank you for catching this typo. The value has now been changed to 236.

“If possible, I'd like to see some more discussion regarding the sex estimations of the individuals in their sample, since this is a key foundation of their study. The authors indicate that they based the sex estimations on the pelvis and skull. It does not look like there were any indeterminate/ambiguous skeletons, which is usually rare. Were all individuals placed into female or male categories based on the pelvis and skull, or was the skull or pelvis used? What if the skull and pelvis “disagreed”?”

Sex was determined mainly on the basis of pelvic morphology, with special attention having been paid to the presence/absence of the ventral arc. Cranial morphology was used to support the pelvic-morphology-based sex determinations. This has now been made clear in the Materials and Methods section—“Both males and females were included in the sample, with sex being determined primarily on the basis of pelvic morphology, especially the presence/absence of the ventral arc [27]. Cranial indicators of sex were also considered when necessary [27].” Lines 89-91.

“A paragraph in the Discussion that addresses the results when pooling females and males reads, to me, more like it belongs in the Results. I suggest it be moved to Results and a brief statement added in the Methods.”

We tried moving the text concerning the combined sex analysis from the Discussion to the Results but it didn’t work well. We found it to be jarring. Consequently, we moved it back to the Discussion.

“I do not see Table 1 that is referenced in the narrative (only Table 2 and Table S1).”

Table 1 has now been moved into the Materials and Methods section (line 93).

“I am curious whether the LDAs were cross-validated. Cross-validation can ensure that the individual under analysis was not included in the method development and is a more accurate reflection of an unknown individual.”

We used the pldam source code in R to run the LDAs, which does include cross-validation. This information has now been added to the Materials and Methods section (lines 173 and 183).

“The last sentence of the Abstract and Conclusion state that "identity in Early Medieval Britain was similar to what it is today--inclusive and dynamic." While this may be true, this study does not examine social identity of past or present peoples (nor the bioarchaeological literature on identity). Ultimately, we do not know how people may have identified (or identified others) and whether such social identities map onto the genetic/ancestral data. I concur that the population was heterogeneous and dynamic in terms of ancestry, but not necessarily in identity. Also, heterogeneity doesn't necessarily equate to "inclusivity." It may be that there were invasions or forced migrations, which arguable isn't inclusion (making space for others). It may also be argued that modern Britain, like anywhere, isn't inclusive. I suggest limiting this conclusion to the heterogeneous and dynamic nature of the ancestries/genetics and avoiding reference to identity altogether.”

Fair point. Both sentences have been changed to “the population of Early Medieval Britain was similar to the population of Britain today—heterogenous.”

*****

Reviewer #3

“In the Materials and Methods section, it is stated that the sample is summarized in Table 1, but there is no Table 1 included in the text. This table should be added to clarify the demographics of the sample. It would be helpful to include date ranges for the Danish sites, as is done with the other sites.”

Table 1 has now been moved into the Methods section (line 93) and includes date ranges for each OTU.

“Each anatomical landmark that was collected should be named and/or described, either within the text, in a new table, or as part of Figure 2.”

A table with the description of each landmark is now available in the supplementary information.

“In Figures 3 and 4, it would be helpful to include the percentage of variance explained along each axis in each of the axis labels.”

The percentages accounted for by each CV have been added to the axes of Figure 3 and 4.

“Would it be possible to describe the variance found along the different CV axes in anatomical terms? What anatomical patterns are described along each axis?”

We chose not to include this information because the actual shape differences between the populations are not important for our results and we felt that including such descriptions would muddy the presentation of the results.

“The second paragraph of the Discussion section is repetitive, using the phrase “our decision to keep the female and male specimens separate did not skew our results” twice. This should be reworded to avoid this repetition.”

The repeated sentence has now been reworded.

“In the seventh paragraph of the Discussion section, there is an inappropriate use of a semicolon, “The historical texts do not mention one; nor do the isotope and DNA studies.” This semicolon should be replaced with a comma.”

The semicolon has been changed to ‘and’.

“It is assumed in the study that variations between the Early and Middle Anglo-Saxon samples are caused by differing patterns of ancestry, but is it possible that there are other factors influencing these differences? Does basicranial morphology change over time in the Danish or pre-Medieval samples?”

It is difficult to entirely discount the possibility that another factor caused the change in shape, but we contend that differing patterns of ancestry is the most likely explanation. First, as we point out in the paper, multiple studies have found that the shape of the basicranium carries a significant genetic signal (Harvati and Weaver 2006; Smith 2009; von Cramon-Taudadel 2009; Smith 2011; Timbrell and Plomp 2019; Plomp et al. 2021). Second, there are two main alternatives to differing patterns of ancestry—natural selection and phenotypic plasticity—and neither of them seems probable. Not only is the time frame short for natural selection, but also there is not an obvious change in environment or behaviour that would have resulted in selection acting on the shape of the basicranium. The absence of an obvious change in environment or behaviour is the primary argument against the phenotypic plasticity hypothesis. That said, we agree that formally exploring the temporal trends in cranial base shape in Britain would be an interesting avenue for future research and we will attempt to get a graduate student to work on the issue in the near future.

“In the Supplementary Document, there are two tables both labeled as Supplementary Table 1. Change the title of one of these tables to Supplementary Table 2. Also, the titles of these tables should be changed to “female” and “male” rather than “females” and “males” since the term “each” is used.”

These have now been changed to Table 3 and Table 4, and the grammar has been corrected.

*****

Reviewer #4

“Table 1 referenced in the first line of the Material and Methods section is absent in the manuscript. If a table is not provided showing sub-sample sizes, including the number of males and females for each group, along with totals, a description of the sample breakdown should be given in the text.”

Table 1 has now been moved to line 93.

“A list of the 34 landmarks used is not given. Page 6, last paragraph refers to landmarks used by Havarti and Weaver (2006) and a figure is referenced in the current study. Once could infer the names of said landmarks, but without labels the reader does not know precisely which landmarks were used. Thus, the study could not be repeated. Include a table listing these landmarks and label the points in Figure 2 with numbers cross-referenced to the new table. A brief summary of how many landmarks are midline and how many are bilateral would be a helpful, as commonly seen in similar studies of bilaterally symmetric structures.”

A table with the description of each landmark is now available in the supplementary information and numbers have been added to Figure 2.

“The observer error sub-study described on page 7 first full paragraph does not indicate whether there was a single or multiple observers. I inferred a single observer and was able to confirm this in the “Authors’ Contributions” section. A mention of a single observer in the text would mitigate expectations for inter-observer error test results.”

A sentence explaining this has been added to the Methods section on line 125.

“The computer program MorphoJ is incorrectly referenced. Dr. Klingenberg’s lab website (https://morphometrics.uk/MorphoJ_page.html) indicates the preferred citation to be Klingenberg, C. P. 2011. MorphoJ: an integrated software package for geometric morphometrics. Molecular Ecology Resources 11: 353-357.”

Thank you for catching this. The reference has been revised accordingly.

“The authors use an uncommon or possibly incorrect symbol for Wilk’s lambda used in reporting MANOVA test results on page 12. Wilk’s lambda is usually abbreviated with the upper-case lambda symbol (Λ) or less commonly a lower-case lambda (λ). The barred lambda given in the manuscript may be a symbol formatting error.”

Thanks for catching this, the incorrect symbols have been replaced with λ.

“The presentation of results is very clear and easy to follow. The authors do not include the effect size for MANOVA results on page 7. Including the partial eta-squared value will help assess the magnitude of the difference in basicranial shape between the groups. Following Cohen J (1992). A power primer. Psych Bull, 112, 155-159, once can interpret partial eta-squared effect sizes as 0.01 = small, 0.06 = medium, and 0.14 = large. A p-value can only tell you if there is a difference or not, but the effect size indicates how large or small that difference is. This may help to tease out subtle differences between groups as statistically significant, but practically insignificant.”

Partial eta scores have now been added to all MANOVAs.

“The two other possibilities for future results on page 19 are well considered. For the first suggestion, the authors should also contemplate including an outgroup in future work to act as a control. Discriminant analysis creates a function that maximizes the differences between groups, no matter how small or practically insignificant they may be. The incorporation of another population removed by geography and/or time can help show that the methodology used is working as it should.”

Thank you for this suggestion. We did not include an outgroup in the current analyses because of how the methods work. For example, if we had included a sample of individuals who are geographically, and thereby genetically, removed from our Northern European individuals, the main variance patterns identified would be those that distinguish the outgroup from the Northern Europeans, while at the same time, the similarities between the Northern European individuals would be highlighted. This would have undermined our ability to identify differences between the Anglo-Saxons and their potential source populations, the Danish and Pre-Medieval Britain. 

“Figures 3 and 4 are well laid out but have axis titles and labels that are difficult to read. The numbers are generally referenced in the text, but the figures are difficult to interpret. Also, the captions for Figures 3 and 4 do not give titles for a or b. ‘a’ is CV1 versus CV2 and ‘b’ is CV2 versus CV3.”

Thank you for pointing out these issues. The font on Figures 3 and 4 have been increased and the captions have been revised.

“The authors interchangeably use the terms “individuals” in place of “specimens” when referring to once living human beings. The connotation of “individual” maintains the humanity of these persons while they provide us with information about our past. Using these terms will help overcome anthropology’s contentious history of digging up the dead without permission.”

We have replaced ‘specimen’ with ‘individual’ throughout the text.

*****

Reviewer #5

“The first CVA computed on the Procrustes residuals of males and females separately probably does not yield reliable results. By my account the number of variables (p*d=102) is 3times larger than any of the groups, which is not in line with general recommendations (e.g. Bookstein & Mitteroecker, 2011).”

We agree. This is the main reason why we opted to perform the LDA on a reduced number of PCs. The CVA plots are provided as a visual representation of the variance but not for statistical analyses.

“The summary of the Manova is confusing: it reports significant differences for pairwise-comparisons and a different value for Wilks' lambda for each of these comparisons. Normally, one would expect one Wilks' lambda for the female Manova and one for the male, so it is not clear what is being reported here and how p-values for pairwise comparisons were obtained.”

We apologise for the confusion here. For each single-sex sample, we performed one MANOVA on the entire sample (four groups) and then six individual pair-wise MANOVAs, thus obtaining a Wilks’ lambda for each comparison. This has now been clarified in the Methods section – “Thereafter, we applied several MANOVAs to the retained PC scores – one that compared all four groups and then six individual pair-wise analyses (e.g. Early Anglo-Saxons vs. Middle Anglo-Saxons, Early Anglo-Saxons vs. Danish, Early Anglo-Saxons vs. Pre-Medieval British, etc.).” Line 159-162

“The key results are obtained by LDA, to classify the Anglo-Saxon samples as either pre-Medieval British or as Danish. The authors find a shift from high to low Danish classification rates between early and Middle Anglo-Saxons. I am a bit skeptical about the female result (only 13 Middle_Anglo-Saxons): It is consistent with the larger male sample and suggests that continental ancestry shifted similarly in males and females. The authors repeated the LDA on a pooled sample to ascertain that the results from the sex-specific analyses are not biased. However, the pooled middle Anglo-Saxon sample remains predominantly male, so some uncertainty remains, as does the possibility of differential ancestry in males and females. Considering that some genetic data suggest high male continental ancestry, it would be worth pursuing this question, though I reckon this would require additional samples from Middle Anglo-Saxon females.”

We agree that further research with a larger sample size would be beneficial, especially if combined with aDNA and isotope data. We hope to pursue this further in the coming years.

As for the current analyses, the CVP PCs were calculated based on balanced samples, which should minimize any error due to sample size differences. This has now been added to Line 159.

While there may have been differences in the numbers of males and females moving to Britain from the Continent during this time period, it is unlikely that our analyses could shed light on this. The only way that our LDA results could be informative about sex differences in ancestry is if we could be certain that every Anglo-Saxon individual identified as having Danish ancestry was a first-generation migrant or if the inheritance of basicranial shape is sex linked. We can’t be certain about the former, and there is no evidence to support the idea that the inheritance of basicranial shape is sex linked. Thus, we contend that the conservative thing to do is to assume that the differences between our female and male LDA results are the result of random variation in preservation and sampling and, therefore, are not meaningful.

That said, the reviewer’s comment did make us think more about the LDA and led us to run a supplementary analysis to ensure that our results weren’t biased. In the additional analysis, we calculated the average contribution of the potential source groups to the two Anglo-Saxon groups. This analysis allowed for the possibility that some of the Anglo-Saxons were the product of intermarriage between locals and non-locals. The results were similar to those found in both the sex-specific and pooled-sex analyses. We have added two new paragraphs in the Discussion (3rd paragraph) outlining these supplementary analyses:

“In the second supplementary analysis, we estimated the contributions of the two potential source groups to the ancestry of the Early and Middle Anglo-Saxons in a different manner. In the main analyses, we used the LDA probabilities to assign each Anglo-Saxon individual to a single potential source group and then calculated the percentage of Anglo-Saxon individuals assigned to each of the two potential source groups. However, it is also possible to treat the LDA percentages as estimates of the contribution of the two potential source groups to each Anglo-Saxon individual’s ancestry and then use the average percentages as the estimates of the contributions of the two potential source groups to the ancestry of Early and Middle Anglo-Saxons. The main difference between these approaches is that the former assumes that the Anglo-Saxons are either locals or non-locals, whereas the latter allows for the possibility that some of the Anglo-Saxons were the products of intermarriage between locals and non-locals.

When the Early Anglo-Saxon females were subjected to the second approach, 69% of their ancestry was estimated to derive from Denmark and 31% from Pre-Medieval Britain. This changed with the Middle Anglo-Saxon females, with 52% of their ancestry being estimated to derive from Pre-Medieval Britain and 48% from Denmark (Supplementary Information Table S3). When the same was done for the Early Anglo-Saxon males, 70% of their ancestry was estimated to derive from Denmark and 30% from Pre-Medieval Britain. As with the females, the pattern changed with the Middle Anglo-Saxon males. Sixty-three percent of their ancestry was estimated to come from Pre-Medieval Britain and 37% from Denmark (Supplementary Information Table S4). Thus, estimating the contributions of the two potential source groups to the ancestry of the Early and Middle Anglo-Saxons in a different way did not change our findings.”

---

## [Decision Letter · Decision Letter 1]

5 May 2021

PONE-D-21-04061R1

A 3D basicranial shape-based assessment of local and continental northwest European ancestry among 5th to 9th century CE Anglo-Saxons

PLOS ONE

Dear Dr. Plomp,

Thank you for submitting your manuscript to PLOS ONE. After careful consideration, we feel that it has merit but does not fully meet PLOS ONE’s publication criteria as it currently stands. Therefore, we invite you to submit a revised version of the manuscript that addresses the points raised during the review process.

There are several minor comments that should be addressed before final acceptance thus I am sending it back to you as a minor revision. Please address the reviewer comments. Note only an editorial review will be conducted upon resubmission.

We look forward to receiving your revised manuscript.

Kind regards,

JJ Cray Jr., Ph.D.

Academic Editor

PLOS ONE

Journal Requirements:

Reviewers' comments:

Reviewer's Responses to Questions

**Comments to the Author**

1. If the authors have adequately addressed your comments raised in a previous round of review and you feel that this manuscript is now acceptable for publication, you may indicate that here to bypass the “Comments to the Author” section, enter your conflict of interest statement in the “Confidential to Editor” section, and submit your "Accept" recommendation.

Reviewer #1: All comments have been addressed

Reviewer #2: (No Response)

Reviewer #3: All comments have been addressed

Reviewer #4: All comments have been addressed

Reviewer #5: (No Response)

2. Is the manuscript technically sound, and do the data support the conclusions?

Reviewer #1: Yes

Reviewer #2: Yes

Reviewer #3: Yes

Reviewer #4: Yes

Reviewer #5: Yes

3. Has the statistical analysis been performed appropriately and rigorously? 

Reviewer #1: Yes

Reviewer #2: Yes

Reviewer #3: Yes

Reviewer #4: Yes

Reviewer #5: Yes

4. Have the authors made all data underlying the findings in their manuscript fully available?

Reviewer #1: Yes

Reviewer #2: Yes

Reviewer #3: Yes

Reviewer #4: Yes

Reviewer #5: Yes

5. Is the manuscript presented in an intelligible fashion and written in standard English?

Reviewer #1: Yes

Reviewer #2: Yes

Reviewer #3: Yes

Reviewer #4: Yes

Reviewer #5: Yes

6. Review Comments to the Author

Reviewer #1: I appreciate the effort the authors made in clarifying the methodological questions I had, as well as the addition of the new analyses. They have successfully addressed all my concerns, and this article will be a meaningful addition to the literature. My only remaining suggestions is to move the discussion of the supplementary analyses to the Results section, instead of the Discussion. I know that one of the other reviewers asked for this and the authors decided not to make these changes, but I'll echo that reviewer's opinion that it does not seems to fit well in the discussion, as it is not truly contributing to the contextualization of the data analyzed.

Reviewer #2: The authors sufficiently incorporated the more substantial issues brought up in the peer review; however, I am unsure if they saw or responded to the PDF'd document I originally uploaded with the first review. I don't see responses to some of the points brought up therein. In particular, I suggest changing "colonization" in the keywords to "ancestry" since the paper doesn't actually discuss colonization, but instead ancestry and population heterogeneity (ln 31); change "men" to "males" since we can't comment on gender as skeletal biologists, and the two terms shouldn't be conflated (ln 62); change "determined" to "estimated" when discussing sex since as skeletal biologists, we cannot determine the sex of someone--we can only estimate their assigned sex (ln 90). There are other minor comments in the PDF that can help with phrasing and clarification.

Overall, the manuscript is improved.

Reviewer #3: Thank you for addressing my concerns and those of the other reviewers.

I have noticed one remaining typo - In line 167, the word "difference" should be used rather than "different."

Reviewer #4: (No Response)

Reviewer #5: The authors have addressed key concerns, thereby clarifying the study design and results, with one exception.

My original comment:

"The authors repeated the LDA on a pooled sample to ascertain that the results from the sex-specific analyses are not biased. However, the pooled middle Anglo-Saxon sample remains predominantly male, so some uncertainty remains, as does the possibility of differential ancestry in males and females. Considering that some genetic data suggest high male continental ancestry, it would be worth pursuing this question, though I reckon this would require additional samples from Middle Anglo-Saxon females."

The authors' response:

"We agree that further research with a larger sample size would be beneficial, especially if combined with aDNA and isotope data. We hope to pursue this further in the coming years. As for the current analyses, the CVP PCs were calculated based on balanced samples, which should minimize any error due to sample size differences. This has now been added to Line 159"

This is not obvious to me: As clearly summarized in the discussion, the pooled LDA was carried out on the entire sample, not on a sex-balanced subsample, so my point remains unaddressed, or something needs to be clarified here. Note the acronym CVP is nowhere defined, I can only guess it refers to the pooled LDA. The reference to Line 159 seems erroneous.

In light of this, I still think that subsample sizes for female Danish and Middle Anglo-Saxons and male early Anglo-Saxons are small relative to the number of variables (PCs retained) and induce uncertainty for the classification results. I am not saying this uncertainty invalidates the interpretation, but it should be acknowledged in the discussion.

7. PLOS authors have the option to publish the peer review history of their article (what does this mean?). If published, this will include your full peer review and any attached files.

Reviewer #1: No

Reviewer #2: No

Reviewer #3: No

Reviewer #4: No

Reviewer #5: No

---

## [Author Response · Author response to Decision Letter 1]

12 May 2021

Responses to reviewers’ comments on the revised Plomp et al.’s ‘A 3D basicranial shape-based assessment of local and continental northwest European ancestry among 5th to 9th century CE Anglo-Saxons’

Our responses to reviewers’ comments are below. Reviewer comments are in black text and our responses are in blue. Thank you again for reading and commenting on our manuscript.

Reviewer #1: I appreciate the effort the authors made in clarifying the methodological questions I had, as well as the addition of the new analyses. They have successfully addressed all my concerns, and this article will be a meaningful addition to the literature. My only remaining suggestions is to move the discussion of the supplementary analyses to the Results section, instead of the Discussion. I know that one of the other reviewers asked for this and the authors decided not to make these changes, but I'll echo that reviewer's opinion that it does not seems to fit well in the discussion, as it is not truly contributing to the contextualization of the data analyzed.

Response: Thank you for the suggestion. We tried moving the supplementary analyses to the Results section, but we found that it did not work as well there as it does in the Discussion. It is also worth noting that the present structure matches that of paper we published in PLOS ONE earlier this year. So, there is precedent for including supplementary analyses in the Discussion in PLOS ONE papers.

Reviewer #2: The authors sufficiently incorporated the more substantial issues brought up in the peer review; however, I am unsure if they saw or responded to the PDF'd document I originally uploaded with the first review. I don't see responses to some of the points brought up therein. In particular, I suggest changing "colonization" in the keywords to "ancestry" since the paper doesn't actually discuss colonization, but instead ancestry and population heterogeneity (ln 31); change "men" to "males" since we can't comment on gender as skeletal biologists, and the two terms shouldn't be conflated (ln 62); change "determined" to "estimated" when discussing sex since as skeletal biologists, we cannot determine the sex of someone--we can only estimate their assigned sex (ln 90). There are other minor comments in the PDF that can help with phrasing and clarification.

Response: We apologise for this oversight. We had missed that there was a PDF with the first round of reviews. We thank the reviewer for taking the time to read the paper so thoroughly. We have edited the text in line with the suggestions in the PDF, with some minor modifications.

Reviewer #3: I have noticed one remaining typo - In line 167, the word "difference" should be used rather than "different."

Response: Well spotted. We’ve made the change.

Reviewer #5: My original comment: "The authors repeated the LDA on a pooled sample to ascertain that the results from the sex-specific analyses are not biased. However, the pooled middle Anglo-Saxon sample remains predominantly male, so some uncertainty remains, as does the possibility of differential ancestry in males and females. Considering that some genetic data suggest high male continental ancestry, it would be worth pursuing this question, though I reckon this would require additional samples from Middle Anglo-Saxon females."

The authors' response:

"We agree that further research with a larger sample size would be beneficial, especially if combined with aDNA and isotope data. We hope to pursue this further in the coming years. As for the current analyses, the CVP PCs were calculated based on balanced samples, which should minimize any error due to sample size differences. This has now been added to Line 159"

This is not obvious to me: As clearly summarized in the discussion, the pooled LDA was carried out on the entire sample, not on a sex-balanced subsample, so my point remains unaddressed, or something needs to be clarified here. Note the acronym CVP is nowhere defined, I can only guess it refers to the pooled LDA. The reference to Line 159 seems erroneous.

In light of this, I still think that subsample sizes for female Danish and Middle Anglo-Saxons and male early Anglo-Saxons are small relative to the number of variables (PCs retained) and induce uncertainty for the classification results. I am not saying this uncertainty invalidates the interpretation, but it should be acknowledged in the discussion.”

Sorry, we should have defined ‘CVP’ in the Materials and Methods. We have now replaced the acronym with ‘the PC-reduction procedure’, which hopefully is clearer.

The number of CVP PCs included in the main analyses did not exceed the smallest group size in each analysis, making them appropriate to use.

We don’t think that the results of our study support the male-bias hypothesis, because the LDA percentages for males and females are pretty similar (Table 3). Nevertheless, we have added some text to the ‘future directions’ part of the Discussion to suggest that potential sex differences in migration and acculturation should be examined with a more comprehensive sample. The relevant paragraph now reads as follows:

“There are two other obvious possibilities for future research. One is to repeat the study with a more comprehensive sample. In particular, it would also be useful to include individuals from regions of Britain not represented in the present study (see Figure 1). This would allow the hypothesis that there was regional variation in the relative number of Anglo-Saxons of continental northwest European and local ancestry to be evaluated. A more comprehensive sample would also allow the possibility that there were sex differences in migration and acculturation to be investigated. In the present study, the results for males and females were similar. However, other studies have suggested that the majority of incomers were male [2].”

---

## [Editor Report · Decision Letter 2]

17 May 2021

A 3D basicranial shape-based assessment of local and continental northwest European ancestry among 5th to 9th century CE Anglo-Saxons

PONE-D-21-04061R2

Dear Dr. Plomp,

We’re pleased to inform you that your manuscript has been judged scientifically suitable for publication and will be formally accepted for publication once it meets all outstanding technical requirements.

Kind regards,

JJ Cray Jr., Ph.D.

Academic Editor

PLOS ONE
---

## [Editor Report · Acceptance letter]

25 May 2021

PONE-D-21-04061R2 

A 3D basicranial shape-based assessment of local and continental northwest European ancestry among 5th to 9th century CE Anglo-Saxons 

Dear Dr. Plomp:

I'm pleased to inform you that your manuscript has been deemed suitable for publication in PLOS ONE. Congratulations! Your manuscript is now with our production department. 

Kind regards, 

on behalf of

Dr. JJ Cray Jr. 

Academic Editor

PLOS ONE